# One-shot to Weakly-Supervised Relation Classification using Language Models

**Thy Thy Tran**[1*]                                  THY.TRAN@MANCHESTER.AC.UK

**Phong Le**[2]                                            LPHONG@AMAZON.COM

**Sophia Ananiadou**[1,3]              SOPHIA.ANANIADOU@MANCHESTER.AC.UK

[1]*National Centre for Text Mining, The University of Manchester, Manchester, United Kingdom*
[2]*Amazon Alexa, Cambridge, UK*
[3]*The Alan Turing Institute, London, United Kingdom*

## Abstract

Relation classification aims at detecting a particular relation type between two entities in text, whose methods mostly requires annotated data. Data annotation is either a manual process for supervised learning, or automated, using knowledge bases for distant learning. Unfortunately, both annotation methodologies are costly and time-consuming since they depend on intensive human labour for annotation or for knowledge base creation. With recent evidence that language models capture some sort of relational facts as knowledge bases, one-shot relation classification using language models has been proposed via matching a given instance against examples. The only requirement is that each relation type is associated with an exemplar. However, the matching approach often yields incorrect predictions. In this work, we propose NoelA, an auto-encoder using a noisy channel, to improve the accuracy by learning from the matching predictions. NoelA outperforms BERT matching and a bootstrapping baseline on TACRED and reWiki80.

## 1. Introduction

Relation classification (RC)[1] detects the connection type between two entities in text such as *place_of_birth* between "Murat Kurnaz" and "Germany", Figure 1. It is crucial for downstream applications such as knowledge base construction [Ji and Grishman, 2011] and question answering [Xu et al., 2016].

Most work in RC relies on either manually- or automatically-annotated data using knowledge bases (KBs) [Zhang et al., 2017, Mintz et al., 2009, Riedel et al., 2010]. This dependency leads to difficulty in generalising such methods to novel domains where labelled data or knowledge bases are not available. Recent work attempts to address the low resource scenario, namely few-shot relation extraction [Han et al., 2018, Baldini Soares et al., 2019], which requires few examples per relation type during testing. However, they rely on a large available training and validation sets to learn the classification task. Perez et al. [2021] suggests to name this setting, *multi-distribution few-shot learning*.

---

[*] This author is currently employed by the Ubiquitous Knowledge Processing (UKP) Lab, Technische Universität Darmstadt. (thytran@ukp.informatik.tu-darmstadt.de)

[1]*Relation classification* is *Relation extraction* with a predefined set of relation types, which assumes a relation hold between the entities.

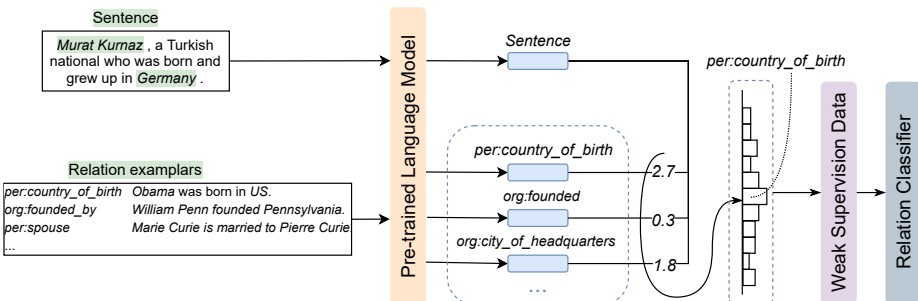

Figure 1: Language models provide weak supervision for relation classification

In contrast, our work evaluates the extreme setting when no separate training and development sets are given. Our setting considers only one example from each relation type, namely *relation exemplar*. We employ the matching model in Baldini Soares et al. [2019] to compute the similarities between an input sentence and those exemplars. After gathering all the scores, we assign the relation with the highest score to the input sentence, such as *country_of_birth* in Figure 1.

The above LM-based one-shot relation predictions are often incorrect. To improve the performance, we propose to learn from these predictions, which we name *noisy annotations*. We propose **NoelA** (short for *Noisy Channel Auto-encoder*), which employs two mechanisms to alleviate the negative impact of noisy labels. Firstly, as entity types have been shown to be helpful for RC [Hancock et al., 2018, Ma et al., 2019, Tran et al., 2020], NoelA reconstructs entity types of the two input entities so that the entity type bias is used when predicting relations. Secondly, we use a noisy channel [Sukhbaatar et al., 2014, Goldberger and Ben-Reuven, 2016, Wang et al., 2019] to explicitly model the prediction noise.

We conducted experiments on two relation classification datasets: TACRED [Zhang et al., 2017] and reWiki80, a modification of Wiki80 [Han et al., 2019]. The two datasets have significantly different characteristics: the relation type distribution (skewed and uniform), the number of relation types (41 and 80), and the domain (news and Wikipedia). We show the performance of different LMs as baselines in order to verify our LM choice. We also demonstrate that bootstrapping [Reed et al., 2014], a traditional approach when supervision is scarce, is not effective. Meanwhile, NoelA outperforms the BERT matching by 9% accuracy on TACRED, and 6% on reWiki80. We conduct an extensive analysis to understand the biases captured in pre-trained LMs and the contributions of each component in NoelA.

## 2. Background

Relation classification (RC) is the task of assigning a (predefined) relation type to a pair of entities in a sentence. We denote $R = \{r_1, r_2, ..., r_m\}$ the set of $m$ relation types. Given a sentence $s$ of $n$ words $s = (w_1, ..., w_n)$, two entities $h, t$ (called head and tail entities respectively), and their corresponding semantic types $e_h, e_t$, the task is to identify the relation $r \in R$ between the two entities. For instance, in Figure 1, $h$ and $t$ are "Murat Kurnaz" and "Germany". Their entity types are $e_h$ = PERSON, $e_t$ = LOCATION. The relation between them is $r = country\_of\_birth$. Our work focuses on predefined relation types excluding NA, i.e. "no relation".

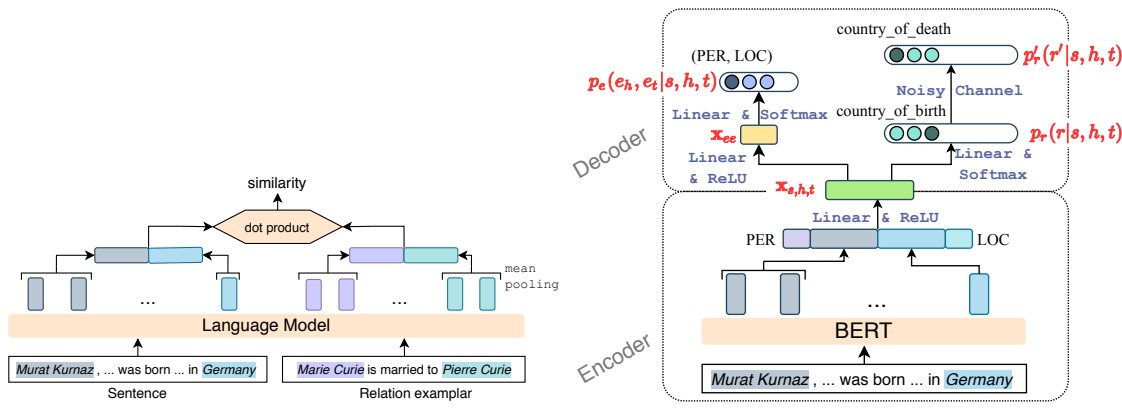

(a) Similarity computation using an LM  (b) The diagram of our model NoelA

Figure 2: (a) Similarity computation using an LM. (b) The diagram of our model NoelA, consisting of an encoder and a decoder. The encoder converts input $\langle s, h, t \rangle$ to a fixed-size vector representation $\mathbf{x}_{s,h,t}$. The decoder then reconstructs the entity types of $h, t$ with $p_e(e_h, e_t | s, h, t)$, and predicts the relation expressed in the input with $p_r(r | s, h, t)$.

A parametric probabilistic relation classifier is given by $p_r(r | s, h, t; \theta)$ (with parameters $\theta$) which assigns a probability to a relation type $r$ given a sentence $s$, head and tail entities $h, t$. Recent RC approaches adopt the diagram $s, h, t \rightarrow$ encoder $\rightarrow$ classifier $\rightarrow r$, converting $\langle s, h, t \rangle$ to a fixed-size vector representation before applying a classification layer (often a pair of linear-softmax over the relation type set $R$)

**True One-shot Setting**   In this work, we assume the extreme scenario where only one-shot training set is available, i.e., a list of desired relation types and a single exemplar for each relation is given. For example, for the relation *country_of_birth*, its exemplar is "[Obama] was born in [the USA]". Those exemplars are short and simple, often less than 10 words. Following Perez et al. [2021], we refer this scenario as *true* one-shot setting to differentiate with previous work [Han et al., 2018], in which a model has access to either separate training or development sets for tuning. Since our work does not aim at creating a new dataset, we use the predefined relation set of the evaluation datasets (TACRED and reWiki80) and manually create an exemplar for each relation type. By creating an exemplar for each relation, we can simulate the process of *imagining artificial examples* when we are given a set of relation types at hand. The relation types and corresponding exemplars used in our experiments are presented in Appendix C.

We note that our one-shot setting is very challenging even without detecting NA (*no relation*). Temporally leaving the problem of predicting NA is in line with the few-shot relation classification research. In particular, the few-shot setting initialised by Han et al. [2018] and further research following it [Baldini Soares et al., 2019] also ignore NA.

## 3. Matching Using Language Models

As shown in recent work, an LM trained on massive raw data (e.g., BERT) can capture some level of semantic similarity. For instance, "[A] is the mother of [B]" is more similar to

"[A] gave birth to [B]" than to "[A] works for [B]". Therefore, we can use matching similarity computed by the LM to assign similar scores between an unseen sentence and the exemplars. Finally, the relation type assigned to the sentence is the one that has the highest score.

Let $f$ be a function mapping sentence $s$ and two entities $h, t$ to a vector in $\mathbb{R}^d$. We compute the matching score $sim(f(s_1, h_1, t_1), f(s_2, h_2, t_2))$, where $sim$ is any function that computes the similarity between two vectors, such as dot product. $f$ produces the vector of $\langle s, h, t \rangle$ in two steps, depicted in Figure 2a. We first compute the entity representations of $h, t$ (i.e., $\mathbf{x}_h, \mathbf{x}_t$) by taking average of the words $(w)$ that the mention contains, e.g., $\mathbf{x}_h = \text{avg}_{w_i \in h}(\mathbf{w}_i)$. Then, we concatenate the two entity representations to form the relation candidate representation $\mathbf{x}_{s,h,t} = [\mathbf{x}_h; \mathbf{x}_t]$. We note that our BERT matching, i.e. when BERT is used, is similar to Baldini Soares et al. [2019]'s BERT with mention pooling.

## 4. Noisy Channel Auto-encoder (NoelA)

Because the prediction by LMs above is inevitably noisy, we propose NoelA (depicted in Figure 2b) to learn from noisy data.

### 4.1 Encoder

The encoder maps $\langle s, h, t \rangle$ to $\mathbf{x}_{s,h,t} \in \mathbb{R}^d$. To compute a vector representation (e.g., $\mathbf{x}_h$) for each entity mention (e.g., $h$), we take mean pooling over the entity span from BERT. We then concatenate the two vectors of $h$ and $t$ with their entity types' embeddings $(\mathbf{x}_{e_h}, \mathbf{x}_{e_t} \in \mathbb{R}^{d_e})$, and apply a linear and a ReLU layers:

$$\mathbf{x}_{s,h,t} = \text{ReLU}(\text{Linear}(\text{Concat}(\mathbf{x}_h, \mathbf{x}_t, \mathbf{x}_{e_h}, \mathbf{x}_{e_t}))) \tag{1}$$

### 4.2 Decoder

Differently from traditional decoders, our decoder does not completely reconstruct the input $\langle s, h, t \rangle$. It reconstructs the entity types $e_h, e_t$ of $h, t$ only, and predicts the relation $r$ hidden in the input by computing $p_r(r|s, h, t)$.

**Relation Classifier**    After having a vector representation of $\langle s, h, t \rangle$, we apply a linear and softmax (over the relation type set $R$) layers to compute $p_r(r|s, h, t)$

$$p_r(.|s, h, t) = \text{Softmax}_R(\text{Linear}(\mathbf{x}_{s,h,t})) \tag{2}$$

**Noisy Channel**    Although a large-scale pre-trained LM is assumed to passively memorise some relational facts, the annotation using such model results in relatively noisy labels. We thus explicitly model the annotation noise by $q(r'|r, s, h, t)$, the probability of assigning $s$ to $r'$ given the correct relation type $r$. This probabilistic function is called "noisy channel" [Goldberger and Ben-Reuven, 2016]. Not knowing the correct $r$, we marginalise over the relation type set $R$:

$$p'_r(r'|s, h, t) = \sum_{r \in R} q(r'|r, s, h, t) p_r(r|s, h, t) \tag{3}$$

It is often assumed that $r'$ is independent from $\langle s, h, t \rangle$ given $r$, thus we base $q(r'|r, s, h, t) = q(r'|r)$ on a matrix $\mathbf{C} \in \mathbb{R}^{|R|^2}$:

$$q(r'|r) = \frac{\exp(c_{r'r})}{\sum_{r''} \exp(c_{r''r})} \tag{4}$$

where $c_{ij}$ is the entry of $\mathbf{C}$ at row $i$, column $j$.

Initialising $q(r'|r)$ has been shown crucial for learning $p(r|s, h, t)$ [Goldberger and Ben-Reuven, 2016]. We initialise $\mathbf{C}$ with a matrix computed by the confusion of choosing relation types by the used LM matching. Formally, let $count(r', r)$ be the number of times when $r' \neq r$ appear together in the top-$k$ candidate lists for all sentences, then

$$c_{r'r} = \log \frac{count(r', r)}{\sum_{r''} count(r'', r)} \tag{5}$$

This initialisation provides to the learning an idea of how much the LM is confused $r$ with $r'$. For instance, *country_of_birth* and *country_of_death* are likely confusing (because in the past an average person often died and was born in the same city/town). On the other hand, *country_of_birth* and *spouse* are easy to distinguish one from the other. In our experiments, we did not fine-tune $q(r'|r)$ and chose $k = \lfloor |R|/4 \rfloor$.

**Entity Type Reconstruction** Another way to tolerate the annotation noise is to inject into the model useful biases. Our encoder uses the entity types of $h, t$ to compute the vector representation $\mathbf{x}_{s,h,t}$, because entity types have been shown to be helpful for RC [Hancock et al., 2018, Ma et al., 2019, Tran et al., 2020]. Intuitively, relation types are constrained by entity types, e.g., *country_of_birth* is constituted by an object such as *person* and a *location*. However, if trained on noisy labels only, the model may not be able to make use of entity types to tolerate the noise. Therefore, we force the model to capture the entity type bias by reconstructing the entity types of $h, t$. Formally speaking, denoting $E$ the entity type set (e.g., $E = \{\text{PER}, \text{LOC}, \text{ORG}, \text{MISC}\}$), we compute the reconstruction probability using a linear and a softmax (over $E \times E$) layers:

$$p_e(.|s, h, t) = \text{Softmax}_{E \times E}(\text{Linear}(\mathbf{x}_{ee})) \tag{6}$$

where $\mathbf{x}_{ee} = \text{ReLU}(\text{Linear}(\mathbf{x}_{s,h,t})) \in \mathbb{R}^{d_{ee}}$.

### 4.3 Learning

Given $\mathcal{D}$ a noisy dataset, we train NoelA by minimising the following loss:

$$L(\theta) = L_{NC}(\theta) + L_{ETR}(\theta) + \lambda L_{DR}(\theta) \tag{7}$$

where $L_{NC}$ is the negative log-likelihood of predicting the noisy labels

$$L_{NC}(\theta) = -\frac{1}{|\mathcal{D}|} \sum_{\langle s, h, t, r' \rangle \in \mathcal{D}} \log p'_r(r'|s, h, t; \theta); \tag{8}$$

$L_{ETR}$ is entity type reconstruction loss, which is the negative log-likelihood of predicting the entity types

$$L_{ETR}(\theta) = -\frac{1}{|\mathcal{D}|} \sum_{\langle s, h, t, r' \rangle \in \mathcal{D}} \log p_e(e_h, e_t|s, h, t; \theta); \tag{9}$$

$L_{DR}$ is the dispersion regularisation term proposed by Simon et al. [2019] and $\lambda \in \mathbb{R}$ is its coefficient. A motivation behind the use of this regulariser is that relation distributions of the noisy data are peaky (Figure 3). Learning from such peaky distributions may lead the model biased towards frequent predicted relations and result in predicting a subset of the relation types. To prevent this issue, we employ the dispersion regulariser that encourages the model to predict a diverse set of relations. Following Simon et al. [2019], we set $\lambda$ to 0.01 in all of our experiments.

## 5. Experiments

Our implementation was developed using the Transformers library [Wolf et al., 2019] and PyTorch [Adam et al., 2017].[2] We use accuracy as evaluation metric. [3]

### 5.1 Settings

**Datasets** We conducted experiments on two English datasets TACRED [Zhang et al., 2017] and reWiki80 whose statistics are shown in Appendix A. TACRED is a widely used dataset for supervised relation extraction, in which we removed the *no relation* instances. reWiki80 is a rearranged variant of the Wiki80 dataset used in Han et al. [2019], originated from FewRel [Han et al., 2018]. Since the test set of Wiki80 is not provided, we used the development set for testing. We took 20% of the training data as the development set for analysis.[4] We tagged the data sets using the Stanford NER tagger [Manning et al., 2014].

The two datasets are significantly different in several aspects. reWiki80 has almost double the relation types than TACRED (80 vs 41). TACRED's relation distribution is skewed while reWiki80's is uniform. TACRED's sentences are from news, and reWiki80's are from Wikipedia.

For each dataset, we manually created a data-agnostic exemplar for each relation in which *head* and *tail* entities were randomly selected and mostly unseen in the two datasets. Considering Wiki80, 51.72% entities in our examplars are seen in the dataset but only 1 pair of entities occur in the training dataset. Regarding TACRED, 28.81% of entities are seen in the dataset but no pairs of entities in the exemplars occur in the training set. We use BERT to generate weak labels on the two training sets, namely *Noisy Data*. The labels originally given in the datasets are *Gold Data*.

**Pre-trained LMs** We examined the base versions of three LMs: BERT [Devlin et al., 2019], GPT2 [Radford et al., 2019], and SpanBERT [Joshi et al., 2020]. The used BERT version is uncased while the used GPT2 and SpanBERT are cased. Note that the BERT matching is similar to the BERT mention pooling model for one-shot RC proposed by Baldini Soares et al. [2019].[5]

---

[2] The source code is available at https://github.com/ttthy/noela.

[3] Accuracy and F1-score are equivalent without a negative class (no relation).

[4] This is acceptable since the gold relations are unseen during training and tuning the model.

[5] We apply mean pooling rather than max pooling because the former outperforms the latter in our preliminary experiments. Although Baldini Soares et al. [2019] show that BERT with entity markers achieves the best performance, it is unclear how the embeddings of entity markers are initialised.

**Relation Classification Settings** The hyper-parameters of our relation classifiers are given in Appendix A. [6] We used the Adam optimiser [Kingma and Ba, 2014] with a widely used learning rate of $3.10^{-4}$, and early stopping based on the *accuracy over the exemplars* with a patience of 5. We note that most of the hyperparameters were taken from previous work as the setting does not allow us to use the gold annotations of the development sets for fine-tuning. Besides, the performance on the exemplar set is not able to reflect the generalisability of a model.

**Models in Comparison** We compare NoelA against several baselines. The first two are **Random** (randomly assigning relation types) and **Frequency** (choosing the most frequent relation types). The others are three LMs (including GPT2-small [Radford et al., 2019], SpanBERT-base [Joshi et al., 2020], BERT-base [Devlin et al., 2019]) and bootstrap-hard [Reed et al., 2014, a bootstrapping approach;]. The bootstrapping model used in our experiments is proposed by Reed et al. [2014], namely *Bootstrap-hard*. The idea is to consider the relation type predicted by the current classifier for an instance at each training step as the noisy label. In particular, we employ the encoder and relation classifier from Section 4 (main text) with the loss $L_{bootstrap-hard}(\theta)$. The loss is a combination of $L_{NC}(\theta)$ and $L_{model}(\theta)$ (Eq. (8) and Eq. (11), respectively), the negative log-likelihood loss of the label predicted by the model with the current $\theta$.

$$L_{bootstrap-hard}(\theta) = \beta L_{nc}(\theta) + (1 - \beta)L_{model}(\theta), \tag{10}$$

where $\beta$ is set to 0.8 following Reed et al. [2014] and $L_{model}$ is computed as follows.

$$L_{model}(\theta) = -\frac{1}{|\mathcal{D}|} \sum_{\substack{\langle s,h,t \rangle \in \mathcal{D} \\ r' = \text{argmax} p(.|s,h,t;\theta)}} \log p(r'|s,h,t;\theta) \tag{11}$$

We also tried their "soft" bootstrapping that minimises the entropy of the predicted label probability distribution $H(p(.|s,h,t;\theta))$. However, the entropy regulariser caused the model collapsed. We thus did not include in our comparison.

### 5.2 Results

The results are shown in Table 1 (further details in Appendix B). In general, the BERT matching yields substantially higher accuracy than the baselines and other two LM matching methods on both datasets. It is worth noting that, the results of BERT in Table 1 are not comparable to those reported in Baldini Soares et al. [2019]. In particular, we take into account all relation types (41 in TACRED and 80 in reWiki), while they consider only 5 or 10 relation types for each testing example ($N$ way $K$ shot setting, §7).

NoelA substantially outperforms bootstrap-hard about 3-6%. Bootstrap-hard only performs on par with BERTwET, our NoelA without the two learning-from-noisy-labels mechanisms and the dispersion regularisation. This suggests that bootstrapping may not work on such noisy data where the seed set is too small, i.e., one example per category.

Removing the components one-by-one reduces the performance of our model. The entity type reconstruction (–ETR) has less impact on the reWiki80 test set, since it only reduces a

---

[6]For a fair comparison with the matching models, we did not fine-tune BERT during training in order to emphasise the contribution of our additional mechanisms instead of the large number of trainable parameters.

|  | **TACRED** | | **reWiki80** | |
|---|---|---|---|---|
|  | Acc. (%) | Abs.+ | Acc. (%) | Abs.+ |
| *Matching* | | | | |
| Random | 2.44 | - | 1.25 | - |
| Frequency | 15.04 | - | 1.25 | - |
| *Pretrained Language Models* | | | | |
| GPT2-small | 0.27 | - | 1.73 | - |
| SpanBERT-base | 8.36 | - | 6.45 | - |
| BERT-base | 15.46 | - | 27.48 | - |
| *Noisy Data* | | | | |
| Bootstrap-hard | 19.28 ±0.42 | 3.82 | 29.76 ±0.16 | 2.28 |
| NoelA | **24.79** ±0.68 | 9.33 | **33.17** ±0.39 | 5.69 |
| –ETR | 21.54 ±0.69 | 6.08 | 32.48 ±0.67 | 5.00 |
| –DR | 21.28 ±0.54 | 5.82 | 32.65 ±0.11 | 5.17 |
| –NC (BERTwET) | 19.03 ±0.34 | 3.57 | 30.06 ±0.14 | 2.58 |
| *Gold Data* | | | | |
| BERTwET (sup.) | 82.73 ±0.99 | 67.27 | 73.92 ±3.46 | 46.44 |

Table 1: Relation classification accuracy (Acc.) of matching baselines, a bootstrapping baseline and our NoelA with its variants. BERTwET (sup.) is the BERT-based classifier without our proposed components, which was trained using the gold relation labels. Except matching baselines, results are average across five runs and the absolute improvement (Abs.+) compared to BERT matching (BERT).

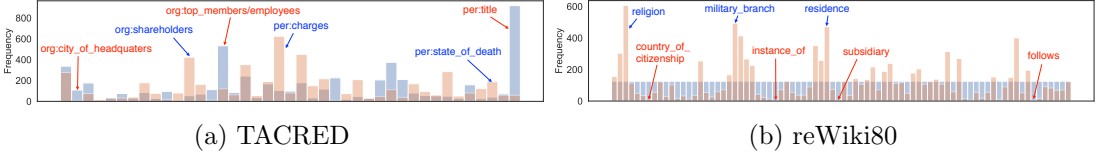

(a) TACRED                    (b) reWiki80

Figure 3: The gold relation distributions and the predicted relation distributions from BERT matching on the development sets. Relation types with high frequency differences between the gold and the predicted distributions are labelled.

marginal score (0.69%). However, the contribution of the dispersion regulariser (–DR) is inconsistent on the two datasets.

## 6. Analysis

### 6.1 Relation Distribution

As the two datasets have different relation distributions, we firstly look at them and those yielded by BERT matching (Figure 3). In TACRED, the gold distribution is skewed towards a few relation types such as *per:title*, *org:top_members/employees*. BERT matching however is in favour of infrequent ones such as *org:shareholders*, *per:charges*. In reWiki80, although the gold distribution is uniform, BERT matching's distribution is multi-modal. This observation shows inappropriate biases of BERT matching, suggesting that one can improve the annotation by injecting inductive bias to BERT matching to make the predicted relation

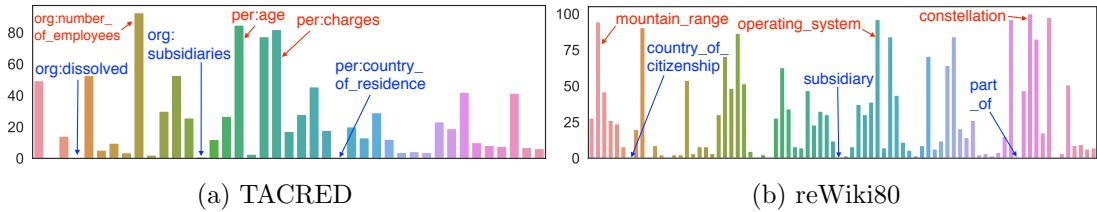

(a) TACRED

(b) reWiki80

Figure 4: Accuracy (%) w.r.t. relation type of BERT matching on the development sets.

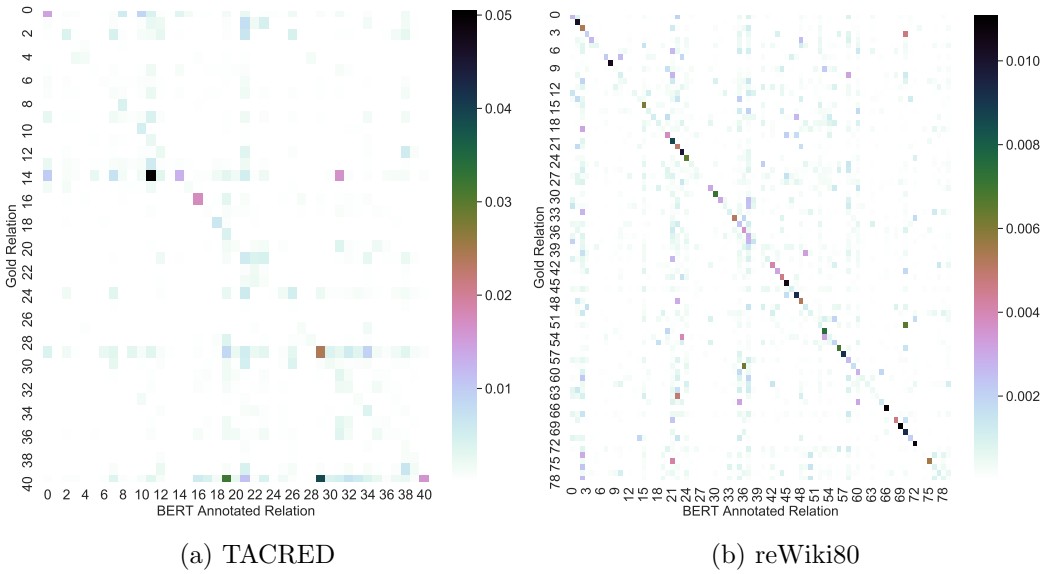

(a) TACRED

(b) reWiki80

Figure 5: Gold relations and BERT matching's relations confusion matrix. The indices of the relation types are given in Table 5 and Table 6.

distribution close to the gold. Furthermore, the difference between the gold distribution and the one yielded by BERT matching by some means explains the under-performance of the bootstrapping baseline as bootstrapping often suffers from semantic drift.

## 6.2 Accuracy of BERT Matching

We show the accuracy of BERT matching according to relation types in Figure 4. On TACRED, BERT matching performs exceptionally well for *per:charges*, *per:age*, but poorly for *org:dissolved*, *org:subsidiaries*. Although Figure 4a gives an intuition that the overall accuracy should be substantially higher than 15.46%, it is not the case because most frequent types have low accuracy. This observation again suggests the need for biasing BERT matching towards frequent relation types. On reWiki80, the highest accuracy is for *mountain_range*, and low is for *part_of*, *subsidiary*, *operating_system*. Because the gold relation distribution is uniform rather than skewed, the overall accuracy of BERT matching on reWiki80 (27.48%) is substantially higher than that on TACRED (15.46%).

BERT matching's confusion matrices are illustrated in Figure 5. The diagonal line is clearly shown for reWiki80 while it is lighter for TACRED. This explains the matching

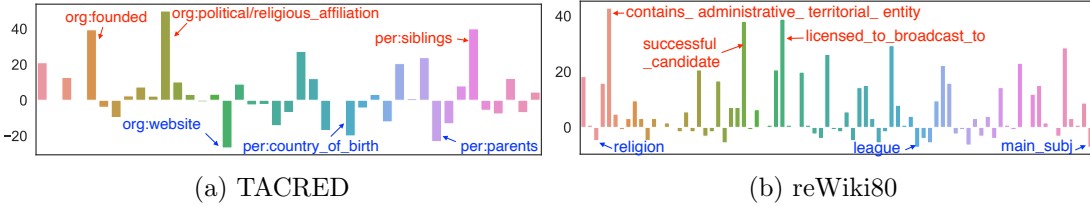

|  (a) TACRED | (b) reWiki80 |

Figure 6:  Accuracy differences (%) w.r.t. relation types between NoelA and BERT matching on the development sets.

performance on TACRED is low, as BERT gets confused by other relations. Generally on both datasets, we observe that BERT matching performs poorly for human-human relations such as parent, mother, sibling, and spouse. The reason is that these relations usually occur in the context of each other. BERT also performs worse on human-location relations such as residence are citizenship that are often hold between the same pair of entities. Meanwhile, place of birth/death are typically the same place in the past, causing the confusion between the two relations.

### 6.3  Accuracy of NoelA versus BERT Matching

Next, we show the accuracy difference of NoelA in comparison with BERT matching in Figure 6. The improvement of NoelA over BERT matching is mediocre on TACRED but visible on reWiki80. Nevertheless, the overall accuracy gain of NoelA on TACRED (9.33%) is substantial. This is due to the skewness of the gold relation distribution of TACRED (Figure 3a): a slight improvement for highly frequent relation types would lead to a substantial overall accuracy improvement.

The observation here indicates an interesting behaviour of NoelA: it seems to adjust its attention according to the hidden gold relation distribution. On TACRED, NoelA trades off the accuracy loss for some infrequent relation types against the accuracy gain for some frequent ones. On reWiki80, NoelA, on the other hand, pays attention to most relation types since all of them are equally frequent.

### 6.4  Impact of Entity Type Reconstruction

We examine to what extend entity types can help to predict gold relations. To do so, we measure the mutual information between entity type pairs (ET) and gold relations (R), and between gold relations (R) and gold relations (R). In information theory, given two random variables $X, Y$, the mutual information $I(X, Y)$ measures the amount of information in $X$ that tells us about $Y$ and vice versa. Therefore, the more helpful to predict relations (R) entity types (ET) are, the larger $I(ET, R)$ is. Because the maximum value of $I(ET, R)$ is $I(R, R)$, we propose the below normalisation

$$\hat{I}(ET; R) = \frac{I(ET; R)}{I(R; R)} \in [0, 1] \tag{12}$$

If entity types are not related to gold relations, $I(ET; R) = 0$; thus $\hat{I}(ET; R) = 0$. Otherwise, if gold relations are determined by entity types, $I(ET; R) = I(R; R)$, leading to $\hat{I}(ET; R) = 1$.

For TACRED $\hat{I}(ET; R) = 0.81$ whereas $\hat{I}(ET; R) = 0.33$ for reWiki80. This explains why the impact of the entity type construction on reWiki80 is substantially smaller than on TACRED.

## 7. Related Work

**Probing pre-trained LMs**  Prior work suggests that pretrained LMs capture factual information that they have seen during pretraining. A few studies have been introduced to extract such factual knowledge from LMs including relations between entities [Petroni et al., 2019, Jiang et al., 2020]. These studies define templates for each relation type, which are used in combination with a given subject to predict the corresponding object. While Petroni et al. [2019] manually define such templates, following studies propose few techniques for automating prompt generation and selection [Jiang et al., 2020, Shin et al., 2020, Gao et al., 2021]. A subsequent line of research is to exploit pre-trained LMs for weak supervision, such as Schick and Schütze [2020] [7] and our two baselines (bootstrap-hard and BERTwET, see Table 1). Our full work goes beyond that by employing learning-from-noisy-label techniques, resulting in substantial improvements.

**Few-shot relation extraction**  Few-shot relation extraction have been introduced firstly by Han et al. [2018], following the $N$ way $K$ shot setting. In particular, $N$ *unseen* relation types and their corresponding $K$ examples are provided during evaluation. This setting assumes access to data from other distributions during training, which allows a model to learn the target task. Moreover, $N$ does not cover the full set of relation types but a subset of it, e.g., $N$ usually equals to 5 or 10. Another setting, requiring less supervision, is the BERT-based one-shot RC, proposed by Baldini Soares et al. [2019]. They evaluate BERT-based relation matching model in $N$ way one-shot setting without using any training data. Their model, however, uses the development set for tuning the hyperparameters. Different from the both settings, our few-shot is *true* in the sense proposed by Perez et al. [2021]: neither training sets or development sets exist.

## 8. Conclusion and Future Work

We demonstrated one-shot relation classification using LMs by generating noisy relation data. To reduce the impact of noisy labels, we proposed **NoelA** (Noisy Channel Auto-encoder) which can learn the latent correct labels by explicitly modeling noise and using entity type bias. NoelA gains a promising 6 and 9% accuracy over BERT on reWiki80 and TACRED, respectively, demonstrating the potential of using weak supervision from LMs. Interestingly, we observed from the analysis of NoelA's accuracy that our model can adjust towards the latent gold relation distribution. We note that *no_relation* is important for relation classification, hence leave it for future work. We also believe that our NoelA is applicable to few-shot because the noisy channel idea is independent on number of examples per relation. In theory, the more examples we have, the better we can denoise the annotation.

---

[7]Because of using an *ensemble* of LMs for less-noisy annotations, this method requires significant more computation and memory than our NoelA. Creating less-noisy annotations like this is orthogonal to our approach, and left for future work.

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

## Appendix A. Experimental Settings

| Dataset | Relation Types | Entity Types | Distribution | Instances | | | Entity Pairs | | |
|---------|---------|---------|---------|---------|---------|---------|---------|---------|---------|
| | | | | Train | Dev | Test | Train | Dev | Test |
| TACRED | 41 | 17 | Skewed | 13,012 | 5,436 | 3,325 | 8,426 | 3,229 | 2,036 |
| reWiki80 | 80 | 8 | Uniform | 50,400 | 10,080 | 5,600 | 50,213 | 10,080 | 5,597 |

Table 2: Data statistics of TACRED and reWiki80 datasets. Each instance is a sentence given entity spans and automatically-labelled entity types.

Table 2 shows the statistics of TACRED and reWiki80 datasets. We used exemplars as the development set and stopped the training process based on the accuracy on it. For every model, we conducted five runs with different initialised parameters and computed the average performance. We list the hyper-parameters of NoelA and bootstrap-hard in Table 3. We note that a small number of instances were eliminated when training the models due to *max length* constraint. The numbers of removed instances in TACRED [Zhang et al., 2017] train/dev/test sets are as follows: 148, 47 and 20 instances, respectively. There is no instance beyond the restricted length in reWiki80 [Han et al., 2019]. Additionally, regarding entity type embeddings, we distinguish the entity types of subject and object, e.g., PERSION-SUBJ and PERSION-OBJ. All experiments were performed on a compute node which has an Intel Skylake CPU and an NVIDIA V100 GPU (16GB GPU RAM).

## Appendix B. Detailed Results

We also report average accuracy over five runs on TACRED and reWiki80 development and test sets in Table 4 with the average training time (Avg. Runtime; minutes). We note that we do not use the development sets during training or for early stopping, we only use them for analysis.

| Parameter | Value |
|---|---|
| Optimiser | Adam |
| Learning rate | 3e-4 |
| Batch size | 128 |
| BERT token dimension | 768 |
| Entity type dimension $d_e$ | 20 |
| Encoder dimension $d$ | 200 |
| Dropout | 0.5 |
| Entity type representation from encoder output $d_{ee}$ | 50 |
| Patience | 5 |
| Max length | 512 |
| $\lambda$ | 0.01 |
| $\beta$ | 0.8 |

Table 3: Hyper-parameters of NoelA and its variants

| | Dev | | Test | | | Avg. Runtime |
|---|---|---|---|---|---|---|
| | Mean | STD | Mean | STD | Abs.+ | |
| | | | **TACRED** | | | |
| Bootstrap-hard | 21.59 | 0.28 | 19.28 | 0.42 | 3.82 | 3.51 |
| **NoelA** | 24.83 | 0.44 | 24.79 | 0.68 | 9.33 | 3.31 |
| –ETR | 21.75 | 0.56 | 21.54 | 0.69 | 6.08 | 2.72 |
| –DR | 20.58 | 1.26 | 21.28 | 0.54 | 5.82 | 2.07 |
| –NC (BERTwET) | 21.97 | 0.18 | 19.03 | 0.34 | 3.57 | 3.26 |
| BERTwET (sup.) | 82.20 | 1.06 | 82.73 | 0.99 | 67.27 | 2.53 |
| | | | **reWiki80** | | | |
| Bootstrap-hard | 30.53 | 0.17 | 29.76 | 0.16 | 2.28 | 4.61 |
| **NoelA** | 33.53 | 0.42 | 33.17 | 0.39 | 5.69 | 4.55 |
| –ETR | 32.88 | 0.44 | 32.48 | 0.67 | 5.00 | 3.88 |
| –DR | 33.00 | 0.25 | 32.65 | 0.11 | 5.17 | 4.59 |
| –NC (BERTwET) | 30.62 | 0.05 | 30.06 | 0.14 | 2.58 | 4.14 |
| BERTwET (sup.) | 79.75 | 4.93 | 73.92 | 3.46 | 46.44 | 3.01 |

Table 4: Average accuracy over five runs on TACRED and reWiki80 development and test sets. We also report the average training time (Avg. Runtime; minutes).

## Appendix C. Exemplars

We present all the exemplars used for TACRED and reWiki80 in Table 5 and Table 6, respectively. All exemplars are manually created by one author and partially revised by another author.

| ID | Relation | Exemplar |
|---|---|---|
| 0 | org:alternate_names | The *World Health Organization* (*WHO*) is a specialized agency of the United Nations responsible for international public health . |
| 1 | org:city_of_headquarters | *Facebook* 's headquarter is located in *Menlo Park, California, United States* . |
| 2 | org:country_of_headquarters | *Facebook* 's headquarter is located in Menlo Park, California, *United States* . |
| 3 | org:dissolved | President Truman dissolved the *O.S.S.* in *1945* . |
| 4 | org:founded | *Facebook* was founded in *2004* . |
| 5 | org:founded_by | *Facebook* was founded by *Mark Zuckerberg* . |
| 6 | org:member_of | *Germany* is a founding member of the *European Union* . |
| 7 | org:members | *Germany* is a founding member of the *European Union* . |
| 8 | org:number_of_employees/members | *IBM* total number of employees in 2019 was *383800* . |
| 9 | org:parents | *Alphabet* is the parent of *Google* . |
| 10 | org:political/religious_affiliation | *Tearfund* is an international *Christian* relief and development agency . |
| 11 | org:shareholders | The largest shareholder of *Google* is *Larry Page* . |
| 12 | org:stateorprovince_of_headquarters | *Facebook* 's headquarter is located in Menlo Park, *California*, United States . |
| 13 | org:subsidiaries | *Cafe Nero* is a child organization of *Rome Bidco* . |
| 14 | org:top_members/employees | *Tedros Adhanom* is the *WHO* current director . |
| 15 | org:website | *gov.uk* is a *United Kingdom public sector information* website . |
| 16 | per:age | *Peter Higgs* is now at the age of *90* . |
| 17 | per:alternate_names | *Mary I of England* was also known as *bloody Mary* . |
| 18 | per:cause_of_death | *Richard Feynman* died of *abdominal cancer* . |
| 19 | per:charges | *Jeffrey Dahmer* was convicted of 15 *murders* . |
| 20 | per:children | *Michael Douglas* is a child of *Kirk Douglas* . |
| 21 | per:cities_of_residence | *Richard Feynman* lived in *New York* . |
| 22 | per:city_of_birth | *Obama* was born in *Honolulu*, Hawaii . |
| 23 | per:city_of_death | *Richard Feynman* died in *Los Angeles* , California , US . |
| 24 | per:countries_of_residence | *Richard Feynman* lived in *US* . |
| 25 | per:country_of_birth | *Obama* was born in the *USA* . |
| 26 | per:country_of_death | *Richard Feynman* died in Los Angeles , California , *US* . |
| 27 | per:date_of_birth | *Obama* was born in *1961* . |
| 28 | per:date_of_death | *Richard Feynman* died in *1988* . |
| 29 | per:employee_of | *Kayleigh McEnany* is the current *White House* press secretary . |
| 30 | per:origin | *Barack Obama* is an *American* politician . |
| 31 | per:other_family | *Craig Robinson* is *Barack Obama* 's brother in law . |
| 32 | per:parents | *Fred Trump* is *Donald Trump* 's father . |
| 33 | per:religion | *Maximilian Kolbe* is *Catholic* . |
| 34 | per:schools_attended | *Peter Higgs* was awarded a PhD degree from *King 's College London* . |
| 35 | per:siblings | *Alexander Watson* is the brother of *Emma Watson* . |
| 36 | per:spouse | *Marie Curie* is married to *Pierre Curie* . |
| 37 | per:stateorprovince_of_birth | *Obama* was born in Honolulu, *Hawaii* . |
| 38 | per:stateorprovince_of_death | *Richard Feynman* died in Los Angeles , *California* , U.S . |
| 39 | per:stateorprovinces_of_residence | *Barack Obama* lives in *Washington* . |
| 40 | per:title | *Barack Obama* was the 44th *president* of the United States . |

Table 5: Exemplars created for each relation in TACRED.

| ID | Relation | Exemplar |
|----|----------|----------|
| 0 | place served by transport hub | *Luton Airport* is an international airport in *London* . |
| 1 | mountain range | The *Tour Noir* is a mountain in the *Mont Blanc massif* . |
| 2 | religion | *Henry VIII* 's religion is *Church of England* . |
| 3 | participating team | *Manchester United* F.C. competes in the *Premier League* . |
| 4 | contains administrative territorial entity | *Ho Chi Minh City* is a territorial entity in *Vietnam* . |
| 5 | head of government | *Barack Obama* is the 44th president of the *United States* . |
| 6 | country of citizenship | *Marco Polo* was an *Italian* explorer . |
| 7 | original network | *One litre of tears* was first aired on *Fuji TV* . |
| 8 | heritage designation | *City of Bath* is listed on *UNESCO World Heritage Site* . |
| 9 | performer | *Abbey Road* is the eleventh studio album by *the Beatles* . |
| 10 | participant of | *Molly Hocking* participated in *The Voice UK 2019* . |
| 11 | position held | *Barack Obama* is the 44th *president of the United States* . |
| 12 | has part | *Germany* is part of *European Union* . |
| 13 | location of formation | *Facebook* was founded in *Massachusetts* . |
| 14 | located on terrain feature | *Heard Island* is located in the *Indian Ocean* . |
| 15 | architect | The architecture of *Eiffel Tower* was designed by *Gustave Eiffel* . |
| 16 | country of origin | *Parasite* is a 2019 *South Korean* black comedy . |
| 17 | publisher | *Harry Potter* was published by *Scholastic* . |
| 18 | director | *Joker* was directed by *Todd Phillips* . |
| 19 | father | *Fred Trump* is *Donald Trump* 's father . |
| 20 | developer | *The Witcher* was developed by *CD Projekt* . |
| 21 | military branch | *Arthur Mackenzie Power* was a *Royal Navy* admiral . |
| 22 | mouth of the watercourse | The *White Nile river Nile* river is a tributary of the Nile . |
| 23 | nominated for | *Spirited Away* was nominated for *Best Animated Feature* . |
| 24 | movement | *Post-impressionist* movement is associated with *Vincent Willem van Gogh* . |
| 25 | successful candidate | *Obama* was elected in *2009* . |
| 26 | followed by | *iPad Air 2* was followed by *iPad Air 3* . |
| 27 | manufacturer | *iPhone* was made by *Foxconn* . |
| 28 | instance of | *Siamese* is a *cat breed* . |
| 29 | after a work by | *Harry Potter and the Cursed Child* is based on a work by *J. K. Rowling* . |
| 30 | member of political party | *David Cameron* was a member of the *Conservative Party* . |
| 31 | licensed to broadcast to | *Tokyo FM* is a radio station in *Chiyoda, Tokyo, Japan* . |
| 32 | headquarters location | *Facebook* 's headquarter is located in *Menlo Park, California, United States* . |
| 33 | sibling | *Alexander Watson* is the brother of *Emma Watson* . |
| 34 | instrument | *Yiruma* plays *piano* . |
| 35 | country | *Corfu* island is in *Greece* . |
| 36 | occupation | *Richard Phillips Feynman* was an American theoretical *physicist* . |
| 37 | residence | *Richard Feynman* lived in *New York* . |
| 38 | work location | *Stephen Hawking* worked in *Cambridge* . |
| 39 | subsidiary | *Cafe Nero* is a child organization of *Rome Bidco* . |
| 40 | participant | *Molly Hocking* participated in *The Voice UK 2019* . |
| 41 | operator | *Stagecoach Manchester* operated the *local bus services* in Greater Manchester . |
| 42 | characters | *Hermione* is a character in *Harry Potter* . |
| 43 | occupant | *Old Trafford Stadium* is occupied by *Manchester United* . |
| 44 | genre | *The Beatles* were an English *rock* band . |
| 45 | operating system | *Microsoft Word* can be installed on *Android* operating system . |
| 46 | owned by | *WhatsApp* is owned by *Facebook* . |
| 47 | platform | *Contra: Rogue Corps* was released for *Playstation 4* . |
| 48 | tributary | The *White Nile river Nile* river is a tributary of the Nile . |
| 49 | winner | *Lara Dutta* was the winner of the *Miss Universe 2000 pageant* . |
| 50 | said to be the same as | *Mary I of England* was also known as *bloody Mary* . |
| 51 | composer | *River flows in you* was written by *Yiruma* . |
| 52 | league | *Alessandro del Piero* plays in *Serie A* league . |
| 53 | record label | *Abbey Road* was released by *Apple Records* . |
| 54 | distributor | *Spirited Away* was released by *Toho* . |
| 55 | screenwriter | *Andrew Lloyd Webber* is the screenwriter of *the phantom of the opera* . |
| 56 | sports season of league or competition | There is a season of *UEFA Champions League* in *2016* . |
| 57 | taxon rank | *Felidae* is a *family* in the taxonomic hierarchy . |
| 58 | location | *The 2008 Summer Olympics* was located in *Beijing* . |
| 59 | field of work | *Alan Turing* was a pioneer of *computer science* . |
| 60 | language of work or name | *Les Miserables* is a *French* historical novel . |
| 61 | applies to jurisdiction | *Mayor of Paris* applies jurisdiction to *Paris* . |
| 62 | notable work | *Vincent van Gogh* is known for *the Starry Night* . |
| 63 | located in the administrative territorial entity | *Ho Chi Minh city* is located in the South of *Vietnam* . |
| 64 | crosses | *Channel Tunnel* crosses *English Channel* . |
| 65 | original language of film or TV show | *Friends* is one of the most-watched *English* language TV shows . |
| 66 | competition class | *Mike Tyson* was a *heavyweight* boxer . |
| 67 | part of | *Netherlands* is part of *Europe* . |
| 68 | sport | *Roger Federer* is a *tennis* player . |
| 69 | constellation | *Andromeda Galaxy* is in the constellation *Andromeda* . |
| 70 | position played on team / speciality | *Cristiano Ronaldo* plays as a *forward* for Juventus . |
| 71 | located in or next to body of water | *Easter Island* is an island in *Pacific Ocean* . |
| 72 | voice type | *Enrico Caruso* has a voice of *tenor* . |
| 73 | follows | *Monday* is after *Sunday* . |
| 74 | spouse | *Marie Curie* is married to *Pierre Curie* . |
| 75 | military rank | *Napoleons* served as a *general* in the French army . |
| 76 | mother | *Marie Curie* is the mother of *Irène Joliot-Curie* . |
| 77 | member of | *Iron Man* is a member of *Avengers* . |
| 78 | child | *Michael Douglas* is a child of *Kirk Douglas* . |
| 79 | main subject | *Robert Langdon* is the main subject of *The Da Vinci Code.* |

Table 6: Exemplars created for each relation in reWiki80.

