# OpenReview forum: "One-shot to Weakly-Supervised Relation Classification using Language Models"
_AKBC.ws/2021/Conference — AKBC 2021_

### Official Review · Reviewer_fjSn · 2021-07-21

**Rating:** 6
**Confidence:** 4

**Review:**

### Summary

This paper proposes a weakly supervised relation classification model using pre-trained language models (LM). It considers the very challenging setting in which only one example of relation type is available. However, LM matching-based predictions which align a test sentence to a relation exemplar, are usually incorrect. This paper proposes to learn from those noisy labeling.

Given a sentence and two entities (and their types), their model first encodes the sentences and then mean pools over the entity span. It then concatenates the entity representation with the representation of their types and passes through a feedforward network (with ReLU). Since the LM-based predictions are usually incorrect, the paper takes a noisy-channel approach in which they model the co-occurrence between the predictions of the noisy and the true label by the model. Since the correct relation is not known, they marginalize over the random variable representing true relation. The intuition behind this loss is that for relations that are easily confused by the model, the loss function increases the probability of the true relations as well (assuming LM-based predictions would be somewhat meaningful and will assign reasonable weight to the ground truth relation as well). Since entity types are helpful for relation classification, the paper has another objective for reconstructing the entity types of the two entities.

The paper conducts experiments on two datasets — TACRED and reWiki80 using multiple LMs (Bert, GPT2, SpanBert). The method achieves modest improvements over the BERT matching system (6% - 9%). However, the absolute performance is pretty low in general (24% for TACRED and 33% for reWiki80). The difference between a model trained with supervision is much higher (> 80% for TACRED and >70% for reWiki80), suggesting that their setup is indeed very hard.

### Strengths:

1. This paper considers a very hard (true one-shot) setting for weakly supervised relation extraction.
2. The noisy channel approach is pretty interesting and has not been applied to few-shot relation extraction before.
3. The paper is well written in general.

### Weakness:

1. Few modeling details are not well motivated (e.g. dispersion regularization term)
2. The bootstrapping baseline should be moved from the appendix to the main section of the paper.

### Question for Authors:

1. Side note: Please number the equations, it makes referring to them easier.
Since the equation for computing the probability of r' marginalizes over all relations, what is stopping from all the p(r | s, h, t) to increase? How do you ensure that the prob of only relations which has high q(r' | r) increase? Would it make sense to also include a term that decreases the entropy of the p(r | h, s, t), i.e. favors peaky distributions?
2. In the loss function, there is a dispersion regularization term. I think the paper lacks the detail regarding why that is required. What would cause the encoder to predict the same relation across all instances?
3. In the end of section 3, you mention a way of encoding the input as [x_h;x_t]. However, later in Sec 4, you redefine how the input is encoded (e.g. ReLU(Linear(Concat(x_h, x_t, x_eh, x_et)))). I am a little confused regarding the initial definition in Sec 3. Is it being used anywhere?
4. "The observation here indicates an interesting behaviour of NoelA: it seems to adjust its
attention according to the hidden gold relation distribution." —- Why do you think this is the case? How is NoelA adjusting performance wrt the gold relation distribution?
5. Suggestion: It would be informative to plot the C (or q(r'|r)) matrix to visualize if BERT also allocates reasonable scores (if not highest) to the correct relation. Edit (I see the confusion matrix is presented in Appendix D, but I think it could make sense to bring it to the main section of the paper.
6. Paper organization suggestion: I think the paper would be more convincing / readable if the analysis in sec 6.1 and 6.2 regarding Bert matching along with the confusion matrix (in Appendix D) is presented to motivate the modeling decision rather than at the end of the paper. Also, please move the details of bootstrapping to the main section of the paper. I am not sure if an additional page is allowed in the camera-ready. If not, to make up for the additional space, I believe the section 6.4 could be moved to the appendix. Similarly size of table 1 could be shortened (fontsize).

---

> ### Author Response · Authors · 2021-07-29
> **response**
>
> We would like to thank the reviewer for the thoughtful comments and questions. We're especially grateful for the writing suggestions (e.g. reconstructing the draft, numbering the equations), and will change our draft accordingly.
>
> **Re the weakness**
>
> > Few modeling details are not well motivated (e.g. dispersion regularization term) (and question 2)
>
> Figure 3 shows that the relation distributions of the noisy data are peaky. Learning from the peaky distributions may lead the model biased towards frequent predicted relations and result in predicting a subset of the relation types. To prevent this issue, we employ the dispersion regularizer that encourages the model to predict a diverse set of relations.
>
>
> **Re the questions,**
>
> > 1. a) What is stopping from all the p(r | s, h, t) to increase?
>
> Because we use a softmax to compute p(r|s,h,t) (please look at "relation classifier", section 4.2), this p is probabilistic. Therefore, when p(.|s,h,t) increases for one r, it has to decrease for some other r's.
>
>
> > 1. b) How do you ensure that the prob of only relations which has high q(r' | r) increase?
>
> Thanks to the backpropagation, if p(r'|r1) > p(r'|r2) then p(r1) increases faster than p(r2) (the conditions s,h,t are temporarily removed). To see that, we can look at the gradient r.t p(r):
>
>     dL / dp(r) = dL / dp'(r') * dp'(r') / dp(r) = dL / dp'(r') * p(r'|r).
>
> Thus, if p(r' | r1) > p(r' | r2), we have |dL /dp(r1)| > |dL/dp(r2)| and hence p(r1) will change faster than p(r2).
>
> When we increase p'(r') as we observe r' in the dataset, we also try to increase p(r) where p(r'|r) > 0 (otherwise, the loss is getting worse). Because |dL /dp(r1)| > |dL/dp(r2)|, p(r1) will increase faster than p(r2).
>
> And now, thanks to the fact that p(r) is probabilistic, p(r) can only increase for some r's whose p(r'|r) are highest.
>
>
> > 1. c) Would it make sense to also include a term that decreases the entropy of the p(r | h, s, t), i.e. favors peaky distributions?
>
> Although decreasing the entropy of p(r) might be helpful, we don't consider that because we don't want to make the model over-confident here. This is because, even without decreasing entropy, neural networks are notorious for being over-confident with their predictions. Being over-confident in our case is extremely dangerous because the training data is noisy. As a result, we are afraid that the model will be more prone to the data noise if we decrease the entropy of p(r).
>
> > 2. In the loss function, there is a dispersion regularization term. I think the paper lacks the detail regarding why that is required. What would cause the encoder to predict the same relation across all instances?
>
> Please see the answer to the weakness above.
>
>
> > 3. In the end of section 3, you mention a way of encoding the input as [x_h;x_t]. However, later in Sec 4, you redefine how the input is encoded (e.g. ReLU(Linear(Concat(x_h, x_t, x_eh, x_et)))). I am a little confused regarding the initial definition in Sec 3. Is it being used anywhere?
>
> We use the encoding [x_h,x_t] for our baselines (e.g. BERT matching, used by Baldini Soares et al. [2019]’s). The encoder described in Section 4 is of our proposed model.
>
> > 4. "The observation here indicates an interesting behaviour of NoelA: it seems to adjust its attention according to the hidden gold relation distribution." —- Why do you think this is the case? How is NoelA adjusting performance wrt the gold relation distribution?
>
> We think this is the case because of the used noisy channel. In the noisy channel, we explicitly model p(r|s,h,t) for the true, hidden relation r. As a result, NoelA might be aware of the fact that the true, hidden distribution of the gold label p(r) is different from the distribution of the noisy label p'(r'). Therefore, NoelA is able to differ p(r) from p'(r'). (Again, s, h, t are temporarily removed for brief.)

---

### Official Review · Reviewer_Rzkz · 2021-07-21
**Solid Contribution to one-shot relation classification**

**Rating:** 6
**Confidence:** 4

**Review:**

This paper considers the problem of one-shot relation classification, in which a model is given a sentence with two entities identified, along with the entity types. The model must predict the label of the relationship, given only unlabeled data and a single labeled example from each possible relationship label. The proposed technique, NoelA, first uses weak supervision by labeling the unlabeled data to train a classifier. Each unlabeled example is embedded with a language model such as BERT and given the label of the nearest neighbor among the labeled examples. Then, the weak labels are used to train a multi-task model. One task head predicts the relationship type. The other is an auto-encoder task that predicts back the entity types of the entities in the sentence. The model also includes dispersion regularization.

Experiments on TACRED and reWiki80 using three language models (BERT, GPT2, and SpanBERT). NoelA offers significant improvements over the baselines of using language models directly for labeling, or training a classifier alone on the weak labels. Ablations also show that the three components of the loss function all contribute to the positive results.

The combination of bootstrapping and multi-task learning is a nice experiment. It's a straightforward pipeline of two common techniques (unless learning to reconstruct entity types is more novel, see question below).

### Questions and comments
- Bootstrapping from one-shot supervision is nice, but how directly tied is it to the benefit of learning to reconstruct entity types. Do any of the references  [Hancock et al., 2018, Ma et al., 2019, Tran et al., 2020] also use an autoencoder component to learn from entity types?
- Something that could strengthen the paper is a discussion of how even a few labeled examples per class could fit into the framework. Some few-shot learning methods can easily handle a variety. Do the authors have an opinion about how best to apply NoelA in the few-shot setting?
- Regarding the claim "Therefore, we force the model to capture the entity type bias by
reconstructing the entity types of h, t." in Section 4.2, is there any way to know that this is really encoding a bias specific to entity types? How do we know, for example, it's not just a generic type of regularization?
- In section 5.1, it says " We manually created a data-agnostic exemplar for each relation in which head and tail entities were randomly selected and mostly unseen in the two datasets." It would be nice to clarify what "mostly unseen" means.

---

> ### Author Response · Authors · 2021-07-29
> **response**
>
> We would like to thank the reviewer for thoughtful comments and questions.
>
> Re the questions
>
>
> > Bootstrapping from one-shot supervision is nice, but how directly tied is it to the benefit of learning to reconstruct entity types. Do any of the references [Hancock et al., 2018, Ma et al., 2019, Tran et al., 2020] also use an autoencoder component to learn from entity types?
>
> Since entity types limit the possible relation types between an entity pair (e.g. "father_of" can only be between two persons), our entity type reconstruction component is proposed to encourage the model taking into account such information. To see how reasonable this is, please refer to section 6.4. Besides, table 1 shows the empirical effect of using entity type reconstruction: when removing the entity type reconstruction component, the performance drops 3.25 and 0.69 % in terms of accuracy on TACRED and reWiki80, respectively.
>
>
> Tran et al (2020) shows that a discrete-state variational autoencoder model reconstructing entity types outperforms much more complicated models taking entity types as input without reconstructing them. Hancock et al., (2018) and Ma et al., (2019) show that entity types impose relation types.
>
> > Do the authors have an opinion about how best to apply NoelA in the few-shot setting?
>
> We believe that our NoelA is applicable to few-shot because the core idea doesn't depend on how many examples per relation. In fact, the more examples we have, the better we can denoise the annotation. We chose one-shot because we aim at minimizing the user's effort in annotation.
>
>
> > Regarding the claim "Therefore, we force the model to capture the entity type bias by reconstructing the entity types of h, t." in Section 4.2, is there any way to know that this is really encoding a bias specific to entity types? How do we know, for example, it's not just a generic type of regularization?
>
> This entity type reconstruction is indeed a type of regularisation as we expect the encoder to behave in some certain way. As the problem of predicting entity types is as difficult as named-entity recognition, the encoder should have some bias specific to entity types.
>
>
> > In section 5.1, it says " We manually created a data-agnostic exemplar for each relation in which head and tail entities were randomly selected and mostly unseen in the two datasets." It would be nice to clarify what "mostly unseen" means.
>
> By "mostly unseen", what we meant is as follows. Considering Wiki80, 51.72% entities are seen in the dataset but only 1.25% (1 pair of entities in the exemplars) occur in the training dataset. Regarding TACRED, 28.81% of entities are seen in the dataset but no pairs of entities in the exemplars occur in the training set.

---

### Official Review · Reviewer_Nx4p · 2021-07-22
**Solid improvement, but I'm skeptical about the setting**

**Rating:** 6
**Confidence:** 4

**Review:**

This paper tackles the semi-supervised relation classification problem with only one manually annotated example for each relation. The method is largely based on the BERT matching model (Baldini Soares et al., 2019). The authors propose to (1) also take entity types as input, (2) use BERT matching to get noisy labels on the unlabeled data, (3) take a noisy channel and an entity type reconstruction objective to train the model on the self-labeled data. The setting is new, so the authors compared their method to BERT matching and a previous bootstrap method. The improvement compared to those selected baseline models is significant, but the performance is in general very low and there is a large gap to the supervised model. There is an ablation study showing the effectiveness of the entity type reconstruction and the noisy channel.

Strength:

The proposed method is very intuitive (noisy channel and self-labeling) and the performance is strong (compared to the baselines).

Weakness:

1. In the experiment, the authors discard all the "no relation" sentences. "No relation" or N/A data are the most challenging part of relation extraction/classification problems, and ignoring them makes it a very toy setting.

2. This one-shot + weakly-supervised setting is not well motivated. First, I didn't quite understand why the authors take the "one-shot" setting instead of taking the "few-shot" setting. Few-shot is in general more applicable in the real world. Second, the authors emphasizes that they are taking a "true few-shot" setting, but they actually take a "toy" weakly supervised setting, where they take the sentences from the supervised dataset, instead of taking an open corpus. Considering the authors also exclude the no-relation instances from the dataset, it makes the setting even easier.

3. This is a minor point: I feel the writing and the structure of the paper could be further improved. The setting is not clearly explained at the beginning.

Question:

1. I'm curious about why the entity type reconstruction can help improve the accuracy. The input already includes the entity types (in the case of TACRED, those are even gold entity types), then why does it matter to "reconstruct" the entity types? Shouldn't the model be able to directly use this information?

---

> ### Author Response · Authors · 2021-07-29
> **response**
>
> We would like to thank the reviewer for thoughtful comments and questions. We're especially grateful for the criticisms, and the writing suggestion.
>
> **Re the weaknesses**
>
> > 1. In the experiment, the authors discard all the "no relation" sentences. "No relation" or N/A data are the most challenging part of relation extraction/classification problems, and ignoring them makes it a very toy setting.
>
> We are fully aware of the importance of NA. However, as our setting is very challenging, we take it step by step. The decision of temporarily ignoring NA is in line with the few-shot relation classification research; for instance, wiki80,  reconstructed from FewRel version 1, ignores NA. BERT matching [Baldini Soares et al. 2019] also ignores it in the few-shot setting. As we mention in our conclusion section, we will tackle the NA prediction problem in our future work (in line with FewRel version 2).
>
>
> > 2. a) I didn't quite understand why the authors take the "one-shot" setting instead of taking the "few-shot" setting.
>
> We believe that our NoelA is applicable to few-shot because the core idea doesn't depend on how many examples per relation. In fact, the more examples we have, the better we can denoise the annotation. We experimented with one-shot in this paper because we aim at minimizing the user's effort in providing examples.
>
>
> > 2. b) The authors emphasize that they are taking a "true few-shot" setting, but they actually take a "toy" weakly supervised setting, where they take the sentences from the supervised dataset, instead of taking an open corpus.
>
> We would like to argue that this is not a "toy" setting because most aspects of our used datasets (except the NA issue which we intentionally left out) are natural. Evaluation on an open corpus shouldn't be more natural because if we manually annotate it, the corpus will be a supervised dataset like the ones we use.
>
>
> Here, we rely on the two supervised datasets because their available manual annotations allow us to analyze the weak labels predicted by BERT, without the need for recruiting human annotators to evaluate the predicted labels.
>
>
> > 2. c) Considering the authors also exclude the no-relation instances from the dataset, it makes the setting even easier.
>
> As BERT matching's performance is very modest on both datasets, 15.46% and 27.48% on TACRED and reWiki80, respectively, we would like to emphasize that our setting is not easy at all. (This is also noticed by reviewer fjSn.)
>
>
> **Re the question**
>
> > The input already includes the entity types (in the case of TACRED, those are even gold entity types), then why does it matter to "reconstruct" the entity types? Shouldn't the model be able to directly use this information?
>
> Although the encoder takes entity types as input, it has a freedom to ignore this information. It is very much similar to what happens to auto-encoder: if we don't reconstruct the input, the encoder has no motivation to "distill" the whole input into a vector.
>
>
> Evidently, Tran et al 2020 shows that a very simple model reconstructing entity types outperforms much more complicated models taking entity types as input without reconstructing them.

---

### Decision · Program_Chairs · 2021-08-17

**Decision:**

Accept

**Comment:**

This paper presents a method for one-shot relation classification. Experimental results demonstrate the effectiveness of the proposed method. The main concern of the reviewers was the limited zero-shot nature of the problem. In spite of that, we think there are enough value in the paper and hence recommend acceptance.